# High-Accuracy Neuro-Navigation with Computer Vision for Frameless Registration and Real-Time Tracking

**DOI:** 10.3390/bioengineering10121401

**Published:** 2023-12-07

**Authors:** Isabella Chiurillo, Raahil M. Sha, Faith C. Robertson, Jian Liu, Jacqueline Li, Hieu Le Mau, Jose M. Amich, William B. Gormley, Roman Stolyarov

**Affiliations:** 1Zeta Surgical Inc., Boston, MA 02111, USA; isabella.chiurillo@zetasurgical.com (I.C.); raahil.sha@zetasurgical.com (R.M.S.); jian.liu@zetasurgical.com (J.L.); jacqueline.li@zetasurgical.com (J.L.); hieu.lemau@zetasurgical.com (H.L.M.); jose.amich@zetasurgical.com (J.M.A.); roman.stolyarov@zetasurgical.com (R.S.); 2Department of Neurological Surgery, Massachusetts General Hospital, Boston, MA 02114, USA; frobertson@mgh.harvard.edu; 3Harvard Medical School, Boston, MA 02115, USA; 4Computational Neurosurgical Outcomes Center, Brigham and Women’s Hospital, Boston, MA 02115, USA; 5Department of Neurological Surgery, Brigham and Women’s Hospital, Boston, MA 02115, USA

**Keywords:** computer vision, image guidance, neuro-navigation, neurosurgery, stereotactic, technology

## Abstract

For the past three decades, neurosurgeons have utilized cranial neuro-navigation systems, bringing millimetric accuracy to operating rooms worldwide. These systems require an operating room team, anesthesia, and, most critically, cranial fixation. As a result, treatments for acute neurosurgical conditions, performed urgently in emergency rooms or intensive care units on awake and non-immobilized patients, have not benefited from traditional neuro-navigation. These emergent procedures are performed freehand, guided only by anatomical landmarks with no navigation, resulting in inaccurate catheter placement and neurological deficits. A rapidly deployable image-guidance technology that offers highly accurate, real-time registration and is capable of tracking awake, moving patients is needed to improve patient safety. The Zeta Cranial Navigation System is currently the only non-fiducial-based, FDA-approved neuro-navigation device that performs real-time registration and continuous patient tracking. To assess this system’s performance, we performed registration and tracking of phantoms and human cadaver heads during controlled motions and various adverse surgical test conditions. As a result, we obtained millimetric or sub-millimetric target and surface registration accuracy. This rapid and accurate frameless neuro-navigation system for mobile subjects can enhance bedside procedure safety and expand the range of interventions performed with high levels of accuracy outside of an operating room.

## 1. Introduction

Global population aging has driven a concomitant rise in neurological conditions that require acute intervention [1]. Pathologies associated with elevated intracranial pressure—stroke, subarachnoid hemorrhage, traumatic brain injury, meningitis, and tumors, all of which can result in acute hydrocephalus—often necessitate urgent or emergent procedures for cerebrospinal fluid (CSF) diversion. One such emergent procedure, an external ventricular drain (EVD), or ventriculostomy, involves the catheterization of CSF-filled intracranial ventricles performed bedside in an emergency room or intensive care unit. EVDs are among the most frequently performed neurosurgical interventions and are considered a gold standard for intracranial pressure monitoring and the method for the treatment of hydrocephalus [2]. However, performing this procedure in an accurate, precise, and efficient manner with non-immobilized patients remains a challenge.

Today, most EVDs are placed freehand using anatomical landmarks, often relying on operator-estimated adjustments based on a patient’s most recent cranial imaging. The freehand approach has remained the standard of care since ventriculostomy procedures were first described in 1744, with an empiric entry location—Kocher’s Point—established in 1894. This method is susceptible to inaccurate catheter placement, often leading to multiple attempts, unnecessary injury to brain parenchyma, and tract hemorrhage, which in turn potentiates catheter occlusion, dysfunction, infection, and, most critically, patient neurologic injury [3,4,5,6]. Inaccurate placement has been cited in approximately 40–50% of procedures performed, with over 20% of catheter tips located off-target [7,8,9,10] and up to 41% resulting in some degree of hemorrhage [11]. As healthcare continuously evolves to reduce risk and improve value, and as the safety of freehand procedures in every area of medicine is being evaluated, it would be clinically beneficial to re-examine the safety of EVD placement. Our group has previously demonstrated that patient-specific entry locations and angles may mitigate the risk of hemorrhage compared to using empiric coordinates, but technology to translate predetermined coordinates and trajectories onto a moving patient outside of an operating room setting has been limited [12].

Existing optical navigation systems are limited to use almost exclusively in the operating room due to their size, complexity, requirement for rigid cranial fixation, and lengthy registration processes. When these devices are occasionally brought to the bedside, setup time can introduce life-threatening delays in critically ill patients. Given the emergent nature of these procedures, neurosurgeons have expressed their unwillingness to adopt technologies that extend EVD procedure time by more than ten minutes [13]. Such limitations have motivated the development of alternative technologies, such as registering pre-procedure images to a fiducial marker that is rigidly attached to a patient [3]. Other approaches utilize electromagnetic systems (e.g., the Stealth Axiem System; Medtronic, Dublin, Ireland) [1], which require secure placement of an electromagnetic control box and manual registration between a volumetric display of radiographic images and user-selected points. However, these approaches fail if the marker is slightly moved or if skin deformation occurs [4].

For the problem of rapid fiducial-less image-to-patient registration, existing algorithms have proven too slow or inaccurate. The heuristics-based method recently presented by Fan et al. has an average surface registration error between 1.28 and 2.24 mm, with multiple seconds of lag between image acquisition and registration [5]. More importantly, average surface error does not represent the true accuracy of a procedure or tool placement, as the target error of a procedure can be up to an order of magnitude greater than the surface error [6]. Though some computer-vision-based algorithms have reported lower surface errors [7,8], these methods require input via handheld scans of the whole patient head, which are time-consuming and require patients to be in a rigid metal frame throughout the scan and subsequent procedure. Additionally, handheld scanners cannot be easily tracked by an instrument tracking system, so these techniques are incompatible with the clinical workflows of an end-to-end navigation platform. Thus, for surface registration to be usable in a surgical setting, it is necessary that the registration system works on a moving patient, is robust to and compatible with clinical workflows, and results in low target registration error rather than surface error, since the former is the only relevant metric to patient safety.

Here, we present an assessment of the Zeta Cranial Navigation System (Zeta Surgical Inc., Boston, MA, USA), an optical, non-fiducial-based neuro-navigation system that performs automatic real-time registration from a patient’s CT scan to live 3D camera images of the patient’s facial anatomy. As the only non-fiducial-based FDA-cleared device that performs real-time registration, this innovation represents a readily deployable system capable of continuous and autonomous correction in the event of patient movement. Our study validates this technology on phantom subjects and human cadaver heads under various challenging test conditions, with the goal that it may be extrapolated to enhance the accuracy and safety of stereotactic cranial procedures.

## 2. Materials and Methods

### 2.1. Zeta Cranial Navigation System

The Zeta Cranial Navigation System used in this study, shown in Figure 1A, was supplied by Zeta Surgical, Inc. The system consists of a Windows computer and associated hardware contained within a cart with two articulating arms. One arm contains a touch-screen monitor. The second arm contains a 3D camera and a position sensor, described in the following sections, rigidly mounted with overlapping capture volumes. The position sensor and 3D camera spaces are calibrated using a tracked board with a specialized printed target, which can be seen by both the position sensor and the 3D camera [14].

The data collected from the system during testing includes transformation matrices describing the positions of the head (phantom or cadaver) and navigated instrument. 

#### 2.1.1. Patient Registration

The Zeta Cranial Navigation System relies on two algorithms to perform continuous patient registration: Snap-Surface and Real-Track. The Snap-Surface algorithm generates a point cloud of the CT image’s surface by segmenting the head CT as a volumetric voxel field using a density-based filter and subsequently converting it into a 3D mesh [15]. This mesh is then converted into a point cloud by randomly sampling triangles on the mesh (weighted by triangle surface area) and then randomly sampling a point within the selected triangle. This sampling process is repeated until 2,016,000 points are selected (Figure 2A,B). The second input is a point cloud of a patient’s face, captured in real-time using a high-accuracy stereo vision 3D camera, specifically the Ensenso N35 (Imaging Development Systems, Inc., Stoneham, MA, USA). This depth image produced by the camera is converted into a point cloud using the pinhole camera model (Figure 2C,D) [16].

Once these point cloud inputs are fed into the system, the Snap-Surface alignment enables an initial, high-fidelity calculation of the transformation matrix that can be used to co-localize the CT data with the patient’s depth image in 3D space. The overall structure of the Snap-Surface algorithm is similar to that of other surface-based registration systems. A feature histogram is calculated for key points in each point cloud, and these feature sets are then matched between clouds to determine point correspondences. These point correspondences are then used to calculate the transformation (Figure 2E–G).

To improve on prior methods, the Snap-Surface algorithm generates a search tree over the set of feature histograms, a space-partitioning search structure that enables a faster search of point correspondences. Using a special search algorithm reduces the required runtime for registration, enabling more key points to be used to generate an accurate initial alignment.

The Real-Track algorithm then enables the initial transformation matrix calculated by Snap-Surface to be updated continuously to account for head movement without compromising registration accuracy. As each subsequent depth image is obtained from the 3D camera, the Real-Track system performs a small local search of the cloud to determine if the head has moved from its prior position (Figure 2H,I). A key component of the Real-Track system is the Iterative Closest Points algorithm (ICP), which uses a gradient descent technique to make minor real-time corrections to the alignment.

#### 2.1.2. Navigated Instrument Tracking

Instrument tracking with the Zeta Cranial Navigation System is performed using a high-accuracy position sensor, the NDI Polaris Vicra (Northern Digital, Inc., Waterloo, ON, Canada), with a volumetric accuracy of 0.25 mm and a 95% confidence interval of 0.5 mm. This position sensor uses near-infrared light to detect and track optical reflective markers on the navigated instrument. For this study, the navigated instrument used was the Brainlab Disposable Stylet (Brainlab AG, Munich, Germany), a catheter stylet with three flat optical markers (Figure 1B).

### 2.2. Phantom Development and Imaging

We developed a rigid 3D-printed PLA phantom of the human head (Figure 3) containing a three-dimensional array of 46 conical divots, allowing repeatable positioning of the Brainlab stylet at known locations. The phantom additionally included four optical markers (NDI Passive Spheres, Northern Digital Inc., Ontario, Canada), intended to track the phantom’s position using a calibrated optical motion capture system (see Section 2.3). 

We obtained two 3D scans of the phantom. The first was a CT scan (EasyTOM industrial 3D scanner, resolution 0.75 mm^3^) intended for use in navigation with the Zeta system. The second was a high-resolution scan obtained from an optical coordinate-measuring machine (Surveyor WS2030 with XLP 500 laser probe, resolution 2 µm^3^) to measure the true divot locations. We then aligned the CT scan and high-resolution scan to the same coordinate system using Volume Graphics version 2022.1 software, allowing direct comparison between instrument positions as measured by the Zeta system and known divot positions from the CMM scan.

### 2.3. Error Calculation

Two key error metrics were calculated to determine the system’s accuracy: target registration error (TRE) and trajectory angle error (TAE). Target registration error (TRE) was determined by placing the tip of the navigated instrument inside a divot on the phantom and calculating the Euclidean distance between the instrument’s tip position as measured by the Zeta system and the position of the divot in the aligned high-resolution 3D model. For any given measurement of TRE under static test conditions, the average, standard deviation, and 99% confidence interval were calculated from 20 samples. For measurements during controlled movement, the navigated instrument was affixed so the tip would stay in the phantom divot, and TRE was determined throughout the movement. 

Trajectory angle error (TAE) was determined using a high-accuracy motion capture system (Optitrack, NaturalPoint Inc., Corvallis, OR, USA), including four mounted PrimeX 41 cameras and the compatible Motive Tracker 2.3.1 software, which has a manufacturer-reported accuracy of 0.1 mm and a 180 Hz frame rate. Trajectory angle error was computed as the angular error between the navigated instrument’s orientation relative to the CT measured by the Zeta system and the motion capture system. For any given measurement of TAE under static test conditions, the average, standard deviation, and 99% confidence interval were calculated from 20 samples. For measurements during controlled movement, the navigated instrument was affixed so it would not move relative to the phantom, and TAE was determined throughout the movement. 

### 2.4. Static Test Conditions

#### 2.4.1. Navigated Instrument Angles

TRE and TAE were measured under various static test conditions, for which the phantom and navigated instrument were rigidly fixed to a test bench, allowing for no movement of either object. Under the first static test condition, TRE and TAE were measured with the phantom frontally facing the camera and the navigated instrument oriented at extreme angles. A total of four orientations were investigated, including a “neutral” position, in which the instrument was frontally facing the Zeta position sensor, and an extreme angle in each rotational dimension, including pitch (the instrument toward/away from Zeta), roll (the instrument rotated across the Zeta field of view), and yaw (the instrument rotated about its axis). Extreme angles were defined as the maximum angles within each dimension at which the Zeta system detected the instrument, and the instrument tip was still inside the phantom divot. After the instrument was set at a given orientation, its rotation from the neutral position was measured using the Optitrack system.

#### 2.4.2. Phantom Angles and Obstruction Conditions

To determine the effect of phantom positioning on registration accuracy, TRE was computed with the phantom rotated from the frontal-facing position and under various obstruction conditions. TRE was measured under the following phantom angles and obstruction conditions: (1) frontally facing the camera under no obstruction; (2) rotated 15° from the frontal facing position; (3) rotated 30° from the frontal facing position; (4) under obstruction of the chin only; (5) under obstruction of the chin, mouth, and nose; (6) under obstruction of the forehead; and (7) under obstruction from placement of an endotracheal (ET) tube. Obstruction conditions are shown in Figure 4.

#### 2.4.3. Phantom Divot Surfaces

To determine whether TRE was different across different parts of the head, TRE was analyzed across 23 divots spanning the five different faces of the phantom (Figure 3). 

#### 2.4.4. Lighting Conditions

TRE was also computed after completing registration of the phantom under three lighting conditions: dark (0 lux), bright (80 kilolux), and maximum illuminance (160 kilolux, corresponding to the maximum central illuminance specified for surgical luminaries in the IEC 60601-2-41:2021 standard [17]). An LED light with a 20,000 Lm luminous flux and 5000 K color temperature was used to illuminate the phantom divot surface, simulating surgical site illumination. To verify the illuminance on the phantom surface, an LX1330B Digital Illuminance Meter was used (Shenzhen Thousandshores Technology Co., Shenzhen, China).

#### 2.4.5. Representative ‘Worst-Case’ Condition

To better predict the system’s performance in clinical settings, TRE and TAE were also measured under a representative ‘worst-case’ test condition. To achieve this set-up, the navigated instrument was oriented at an extreme angle in each rotational dimension, the phantom was rotated 30° from the frontal-facing position, and the phantom was obstructed so that the forehead was only partially visible and below the nose was fully covered.

### 2.5. Movement Conditions

To assess the system’s ability to maintain high registration accuracy during motion, TRE and TAE were also measured under controlled movements of the phantom. The phantom was manually translated across the field of view of the 3D camera at varying speeds, with the navigated instrument rigidly affixed and facing the position sensor. The phantom’s linear speed throughout the entire motion was calculated from position data measured with the Optitrack motion capture system, and for all measurements of TRE and TAE, the phantom’s speed at that time was interpolated from this data. We computed the maximum speed at which the system was able to maintain TRE under 2 mm, which is the generally accepted threshold for clinical use of neuro-navigation systems today [18]. This maximum speed was computed under an optimal test condition (with the phantom frontally facing the camera and the navigated instrument frontally facing the sensor) and under a representative ‘worst-case’ test condition (described in Section 2.4.5 Representative ‘worst-case’ condition).

### 2.6. Registration and Instrument Tracking Latency

Registration latency was defined as the period between the following two points: (1) the moment at which a head already registered with the Zeta system’s Snap-Surface alignment system is moved, and (2) when the Real-Track alignment system accurately reports the head’s positioning. Similarly, instrument tracking latency was defined as the period between the following points: (1) the moment at which a navigated instrument is moved and (2) when the Zeta system accurately reports the instrument’s new position. 

The phantom and a rigidly affixed navigated instrument were positioned frontally, facing the camera and position sensor. This assembly was manually rotated in an oscillatory motion so the navigated instrument would continuously be in view of the position sensor while the net angular displacement profiles of the phantom and instrument were measured with the Optitrack system. The registration and instrument tracking latency were computed by performing a cross-correlation of the assembly’s net angular displacement signals as measured by, respectively, the Zeta system and the Optitrack motion capture system. To compute the cross-correlation, the measurements made with the registration and instrument tracking systems were upsampled to match the frequency of the Optitrack motion capture system, 180 fps.

### 2.7. Cadaveric Preparation, Imaging, and Analysis

We performed cadaveric validation of the Zeta system using five cadaveric heads. The specimens were preserved by freezing and later thawed to their original state before testing. The cadavers spanned different skin tones (four white, one black), sexes (three male, two female), and head sizes (covering a range of 150–165 mm width). CT scans were obtained for each cadaver of the entire head without gantry tilt or contrast at a slice thickness of 0.5 mm with a 512 × 512 resolution.

Given the inherent challenges in establishing a reliable ‘ground truth’ for instrument position on cadavers (like phantom’s divot positions obtained from a high-resolution scan), the cadavers were used to evaluate the repeatability of navigated instrument position and trajectory angle rather than absolute TRE and TAE. For each cadaver, twenty targets were marked on the back surface of the head, and a small incision was made so the navigated instrument could be rigidly positioned with the tip on each target. After completing registration for the cadaver, 20 measurements of instrument position and trajectory angle were taken for each target.

The variability in these repeated measurements for each target was quantified using the standard deviation of instrument position and trajectory angle. Subsequently, to provide a summary metric of the registration consistency with cadavers, the average standard deviation was computed across all targets for instrument position and trajectory angle, referred to here as instrument position variability (IPV) and instrument angle variability (IAV), respectively. For comparison, the IPV and IAV were also computed for 20 targets on the phantom. These metrics gauge the registration and instrument tracking systems’ repeatability without reference to a fixed ground truth.

### 2.8. Statistical Analysis

All statistical analyses and plotting were completed using Python and the Matplotlib library (Copyright, John D. Hunter). The resulting errors of the various test conditions were compared to their baseline values using pairwise Student’s independent *t*-tests.

## 3. Results

### 3.1. Target Registration Error and Trajectory Angle Error under Static Test Conditions

Figure 5A shows TRE and TAE across extreme instrument angles. At a neutral angle, we measured a TRE of 0.926 ± 0.176 mm and a TAE of 0.429 ± 0.067°. An increase in mean TRE was observed for both maximum pitch and roll angles, and an increase in mean TAE was observed for maximum yaw and roll angles.

Figure 5B shows TRE across phantom angles and obstruction conditions. The mean TRE spanned from 0.792 mm to 1.236 mm across all obstruction conditions, with the highest error for complete upper obstruction.

Figure 5C shows TRE across all phantom divots. The mean TRE ranged from 0.712 mm to 1.026 mm, and no statistically significant difference in TRE was observed across the five divot faces of the phantom (see Figure 5D for corresponding divot locations).

Lighting conditions did not significantly affect TRE, with errors of 0.804 ± 0.113 mm, 0.815 ± 0.124 mm, and 0.795 ± 0.112 mm for the dark, bright, and maximum illuminance conditions tested, respectively.

Finally, under worst-case conditions, which included a combination of extreme navigated instrument angle (54.197° pitch, 18.125° roll, and 13.506° yaw), 30° phantom angle, and phantom obstruction, TRE and TAE were respectively observed to be 1.0378 ± 0.227 mm and 0.496 ± 0.080°.

### 3.2. Target Registration Error and Trajectory Angle Error during Movement

We calculated the phantom’s linear speed throughout the entire motion from position data measured with the Optitrack motion capture system (Figure 6A), and for all measurements of TRE and TAE, we interpolated the phantom’s speed at that time from this data (Figure 6B). To identify the maximum speed at which the system maintains high accuracy, for any given speed, *v*, we computed the mean, standard deviation, and 99% confidence interval of TAE and TRE from samples taken at speeds in the range [*v*, *v* + 0.35 cm/s] (Figure 6C). This range allowed for each computation to include at least 20 samples of TRE and TAE.

With the phantom frontally facing the camera and the navigated instrument frontally facing the sensor, the Zeta system maintained TRE below 2 mm for all speeds up to 1.773 cm/s, at which point the upper bound 99% confidence interval of TRE exceeds the 2 mm bound (Figure 6C). The mean TAE across all speeds tested, up to 2.015 cm/s, was 0.322 ± 0.015°. Under the representative ‘worst-case’ test condition, the maximum speed for which the Zeta system maintained TRE below 2 mm was 0.875 cm/s, and the mean TAE across all speeds tested, up to 0.966 cm/s, was 0.761 ± 0.017°.

### 3.3. Registration and Instrument Tracking Latency

To determine the registration and instrument tracking systems’ latencies, we performed a cross-correlation between the phantom’s and instrument’s net angular displacement during oscillatory motion as measured by the Zeta system and the Optitrack motion capture system. This computation resulted in a registration latency of 0.167 s and an instrument tracking latency of about 0.0 s.

### 3.4. Cadaveric Experimental Results

To better understand the Zeta system’s performance in a clinical setting, instrument position variability (IPV) and instrument trajectory angle variability (IAV) were measured for the phantom and across five cadaveric heads. As shown in Figure 7, the mean IPV for the phantom head was 0.067 ± 0.014 mm, and the mean IAV was 0.052 ± 0.075°. As expected, IPV and IAV increased across the cadavers, at 0.219 ± 0.079 mm and 0.192 ± 0.147°, respectively.

## 4. Discussion

### 4.1. Registration Accuracy under Static Conditions

Millimeter or sub-millimeter accuracy was achieved using the Zeta system across all static test conditions. Although increases in TRE and TAE were observed for some test conditions, TRE was significantly below 2 mm across all results, which is the generally accepted threshold for neuro-navigation systems. Compared to the baseline (with the phantom frontally facing the 3D camera and the navigated instrument frontally facing the position sensor), an increase in TRE and TAE was observed when the navigated instrument was oriented at extreme angles. This is likely due to the navigated instrument having flat optical markers, which are detected best by the position sensor when facing neutrally toward it.

Similarly, compared to a baseline measurement (with the phantom frontally facing the camera under no obstruction), TRE was found to increase when key features of the phantom were significantly obstructed (i.e., when the phantom’s forehead was entirely obstructed or when the phantom was rotated 30° away from the camera). This effect was not observed in the other angles and conditions, likely due to the registration being accurate enough to make the effect negligible.

Although no statistically significant difference in TRE was observed across the five divot faces of the phantom, compared to the overall average TRE, a slightly higher TRE was measured at some divots. Since TRE for all targets was computed after a single Snap-Surface registration, it is more likely that the differences in TRE observed are due to variability in the placement of the navigated instrument tip into the divot rather than the target’s location on the phantom.

Most notably, the Zeta system maintained millimeter accuracy under a representative worst-case test condition in which multiple variables, including instrument angle, phantom angle, and phantom obstruction, were tested. 

### 4.2. Registration Accuracy during Movement

Testing of the Zeta system revealed that TRE could be maintained below 2 mm with the phantom and the navigated instrument in motion. Specifically, under an optimal test condition, TRE was maintained below 2 mm, with the phantom being translated at speeds up to 1.773 cm/s. Under a representative ‘worst-case’ test condition, TRE was maintained below 2 mm up to a speed of 0.875 cm/s. We expect typical system performance to vary between these values, influenced by factors such as the head angle and/or obstruction of key features. Furthermore, the registration latency computed was only 167 ms, while the instrument tracking latency was about 0.0 s (indicating instrument updates are almost instantaneous). These results indicate that the Real-Track algorithm can maintain high-accuracy registration even when the head is moving. 

### 4.3. Cadaveric Validation

Cadaveric validation of the system was performed by computing the IPV and IAV for both the phantom and across five cadaveric heads. Although IPV and IAV were found to be larger in the cadaveric tests, this does not directly correlate to a proportional increase in TRE and TAE. Given that the system demonstrated low TRE and TAE on the phantom across variable test conditions, we anticipate that the system should maintain a similar level of accuracy on cadavers despite the heightened variability.

### 4.4. Advantages of the Proposed Method

Image-guided software with accurate and efficient registration for mobile, non-fixed subjects could substantially improve patient safety in procedural medicine, particularly in neurosurgery, where one of the most common procedures is freehand ventriculostomy. The Zeta Cranial Navigation System utilizes computer vision technology to fuse patients’ computed tomography scans to live 3D camera images (Snap-Surface) with automatic tracking of patient movement (Real-Track). To our knowledge, this is the first study to demonstrate an automatic, sub-millimetric facial registration and tracking device that can follow patients in motion. By evaluating several test conditions, including facing the head towards the 3D camera at different angles, partially draping the head to obstruct various facial features, and under various illumination conditions, we simulated challenging hospital settings to facilitate the translation of this tool into patient care. 

Not only would this approach facilitate the safe placement of EVD catheters, eliminating neurologic injury caused by multiple failed attempted procedures and the resulting tissue damage, but the enhanced reliability of precise and accurate placement could increase opportunities for subsequent intra-ventricular and intracranial interventions. For example, in cases of intraventricular thrombolytic therapy where a recombinant tissue plasminogen activator was used to treat intraventricular hemorrhage, patients incurred a 23% tract hemorrhage rate, due largely to the fact that 30% of catheters did not terminate in the desired location [19,20]. Results such as these demonstrate that the absence of techniques for ensuring reliable EVD placement can pose limitations on the ability to provide necessary patient care [21]. Given more accurate catheter placement, one could envision surgery through the same EVD channel without the need for any further injury to the brain parenchyma. Furthermore, this technology could facilitate interventions such as stereotactic needle biopsies and electroencephalography electrode placement or be extrapolated to other procedural fields facing a similar challenge of image registration on mobile patients.

The application of sub-millimetric accuracy neuro-navigation on awake patients without the use of cranial fixation also opens the use of this technology to new, promising, non-surgical therapeutic cranial interventions that have suffered from a lack of accurate and mobile neuro-navigation. These new procedures often require multiple applications of therapy on patients who are outpatients and for whom anesthesia and cranial fixation would represent large impediments to wide-scale adoption. These procedures include transcranial magnetic stimulation (TMS) for diseases such as depression and focused ultrasound for intracranial drug delivery and stimulation-based treatment of functional disorders including addiction, depression, and obesity [22,23,24]. The ability to perform neuronavigation on awake, non-immobilized patients has the potential to make these new procedures more accessible and affordable.

The previously mentioned advantages of the system also have the potential to reduce healthcare costs. For instance, facilitating the accurate placement of EVDs on the first attempt can reduce the diagnostic, procedure, and material costs associated with complications and catheter replacements. Furthermore, the Zeta system’s independence from cranial fixation expands its usability beyond conventional operating rooms. This adaptability enables certain procedures to be conducted in a standard procedure room, which can reduce the expenses associated with operating rooms such as anesthesia, specialized equipment, and staffing.

Despite the many advantages described, the study’s limitations warrant further discussion. This study did not assess the system’s robustness in accounting for deformations of the patient’s face that could occur between the time when the CT was obtained and when registration was attempted. As this study only utilized a rigid phantom for computing target registration error and trajectory angle error, a potential avenue for future work could include using a calibrated phantom with a deformable face to compute these error metrics. Even so, it is not expected that the system could perform with high accuracy if there is a significant difference between the preoperative scan and the intraoperative point cloud, such as if an outdated scan is used or if there is an extreme amount of swelling after the patient is scanned. Additionally, it would be of interest to assess the system’s accuracy using phantoms with different skin textures, skin tones, and facial hair. Finally, it would be interesting to investigate the performance of this system using phantoms of different head sizes or while varying the distance between the 3D camera and the phantom, as this could affect the density and size of the point cloud used for registration, and consequently the computational load and overall performance of the registration algorithms.

## 5. Conclusions

Despite tremendous advances in modern image-based navigation technologies utilized in the operating room, neuro-navigation for bedside cranial procedures is limited. We present a novel device, the Zeta Cranial Navigation System, that uniquely combines a registration and real-time tracking system without reliance on cranial fixation, fiducials, or other markers. This will allow for the performance of neuronavigation in awake, non-immobilized patients—an ability with far-reaching consequences, including how and where intracranial procedures can be performed. Furthermore, the automated registration provides millimeter or sub-millimeter accuracy, and the motion tracking operates in real time; both variables are essential for ensuring patient safety during intracranial procedures, with tremendous potential for improving the safety of neurosurgical care while expanding the range of interventions that can be performed outside of an operating room.

## Figures and Tables

**Figure 1 bioengineering-10-01401-f001:**
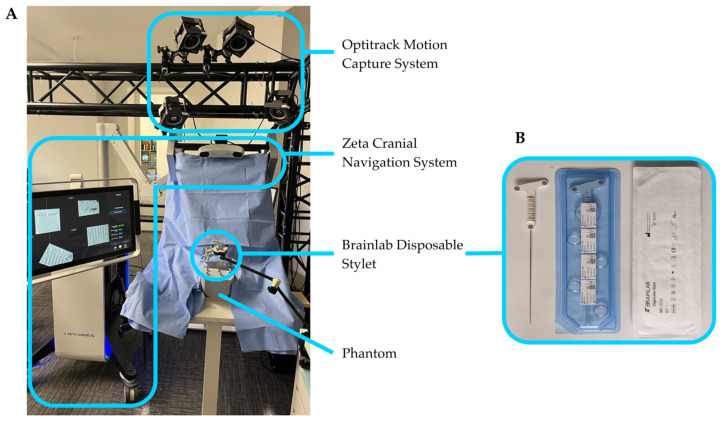
(**A**) Experimental setup employed in this study: Zeta Cranial Navigation System, Optitrack Motion Capture System, Brainlab Disposable Stylet, and phantom. (**B**) Close-up view of the Brainlab Disposable Stylet.

**Figure 2 bioengineering-10-01401-f002:**
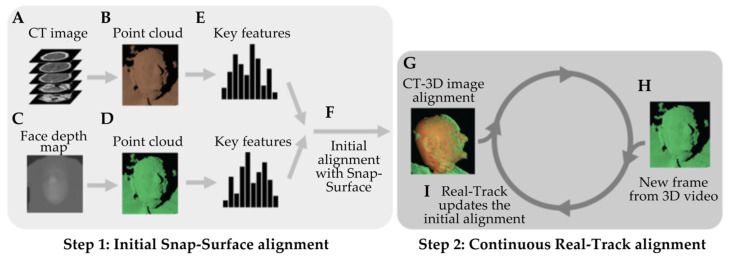
Registration system overview: (**A**) A preoperative CT or MRI is obtained for the patient. (**B**) The patient’s preoperative facial map is segmented from CT or MRI data using a density-based filter to visualize regions with similar properties. The facial map is subsequently converted into a point cloud. (**C**) A 3D camera is used to image the surface of the patient’s face to obtain an intraoperative facial depth map. (**D**) The intraoperative facial map is subsequently converted into a point cloud. (**E**) Key features are extracted from both point clouds. (**F**) The Snap-Surface alignment algorithm registers both point clouds, generating a transformation matrix that allows the co-localization of CT or MRI data with the patient’s 3D image. In other words, the CT or MRI image is mapped to the surface of the patient’s face visible to the 3D camera. (**G**) An initial alignment between the CT or MRI and the patient is obtained. (**H**) The 3D camera continuously captures new 3D images of the patient’s face, which are similarly converted to intraoperative point clouds. (**I**) The Real-Track algorithm uses the new intraoperative point cloud and updates the initial registration. This process is repeated throughout the procedure at a rate of about five frames per second.

**Figure 3 bioengineering-10-01401-f003:**
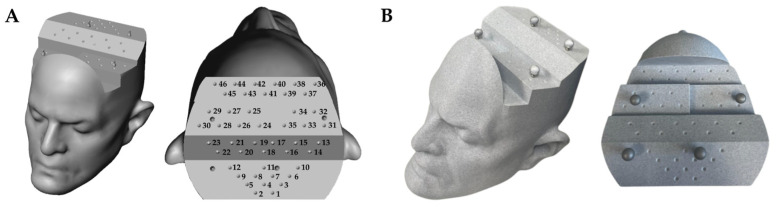
(**A**) Phantom 3D model with 46 marked conical divots arrayed on five faces. (**B**) 3D-printed phantom.

**Figure 4 bioengineering-10-01401-f004:**
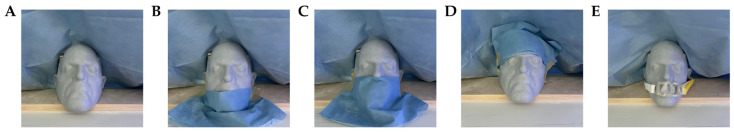
Phantom obstruction conditions: (**A**) no obstruction; (**B**) obstruction of the chin only; (**C**) obstruction of the chin, mouth, and nose; (**D**) obstruction of the forehead; and (**E**) obstruction from placement of an endotracheal (ET) tube.

**Figure 5 bioengineering-10-01401-f005:**
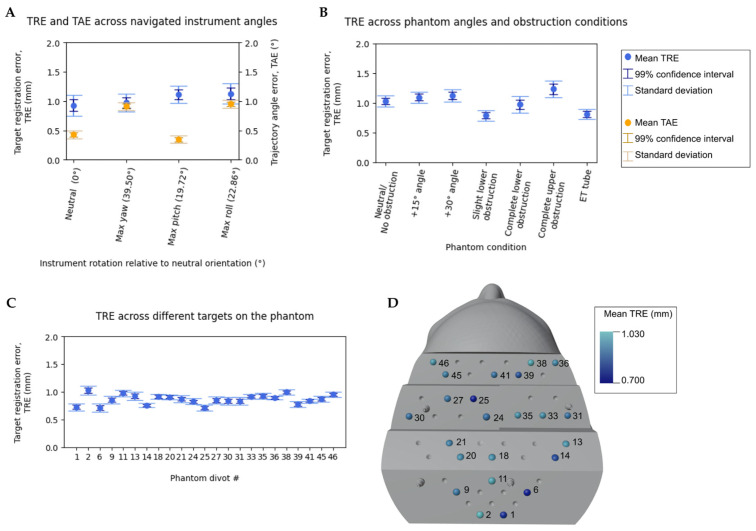
The mean, standard deviation, and 99% confidence interval for TRE and TAE were measured under various static test conditions. (**A**) TRE and TAE when the navigated instrument is rotated to extreme pitch, roll, and yaw angles. (**B**) TRE across two phantom angles and four obstruction conditions. (**C**,**D**) TRE across 23 phantom divots, spanning all five faces of the phantom.

**Figure 6 bioengineering-10-01401-f006:**
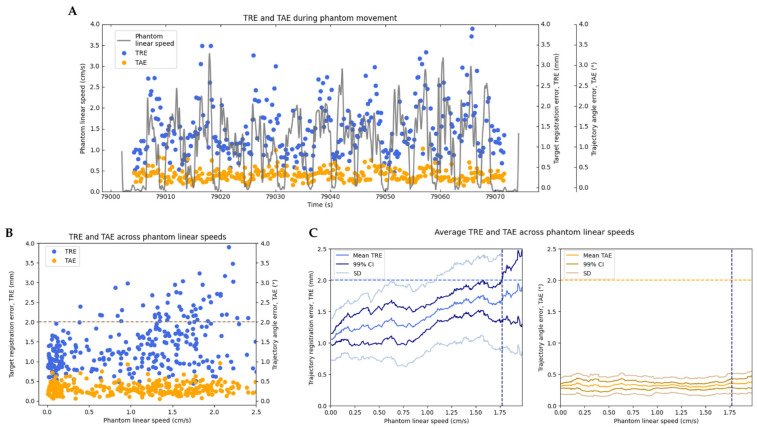
Registration accuracy during phantom movement. (**A**) TRE and TAE computed during phantom movement. (**B**) TRE and TAE across measured phantom linear speeds. (**C**) Average TRE (**left**) and TAE (**right**) across phantom linear speeds. The horizontal dotted lines represent the maximum acceptable TRE and TAE (at 2 mm and 2°), and the vertical dotted lines indicate the maximum phantom speed for which the 99% confidence interval of both TRE and TAE are maintained below the 2mm and 2° threshold.

**Figure 7 bioengineering-10-01401-f007:**
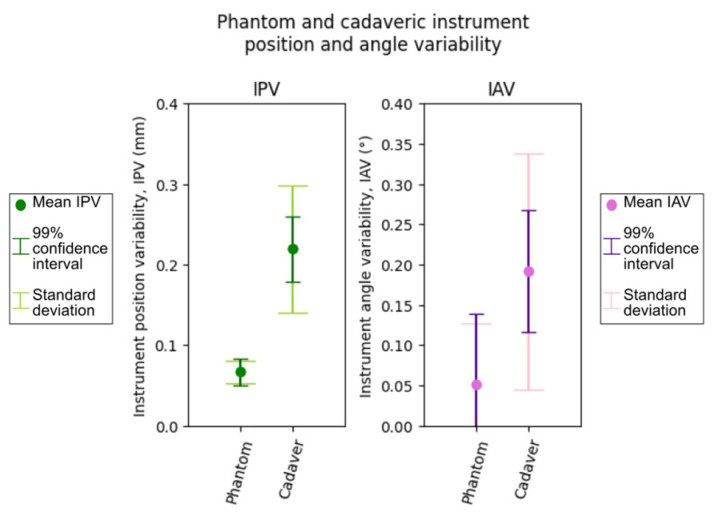
The mean, standard deviation, and 99% confidence interval for phantom and cadaveric instrument position variability (IPV) and instrument angle variability (IAV). Phantom IPV and IAV were computed for 20 targets on the phantom. Cadaveric IPV and IAV were computed for 20 targets each across five cadavers, for a total of 100 data points.

## Data Availability

The data presented in this study are available on request from the corresponding author. The data are not publicly available due to privacy reasons.

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
