# Peer review of "High-Accuracy Neuro-Navigation with Computer Vision for Frameless Registration and Real-Time Tracking"

_bioengineering, 2023, doi:10.3390/bioengineering10121401_

Round 1
Reviewer 1 Report
Comments and Suggestions for Authors
The authors present an assessment of the Zeta Cranial Navigation System, an optical, non-fiducial-based neuro-navigation system that performs automatic real-time registration from a patient's CT scan to live 3D camera images of the patient’s facial anatomy. Overall, This innovation represents a readily deployable system, capable of continuous and autonomous correction in the event of patient movement, but still needs major revisions before it can be considered for publication.
1. As a simple procedure for"Non-neurosurgeons performing emergency life-saving procedures", the navigation system should be more visible during the operation of the interface, so that readers have a more direct and complete understanding of the system;
2. Instrument position variability (IPV) and instrument trajectory angle variability (IAV) of the prosthesis and 5 cadaver heads should be shown in scatter plots for more intuitive and accurate comparisons;
3. The investigation into the effects of motion conditions on tracking accuracy and instrument delays is too modest to add a broader range of variables.
4. In this paper, the accuracy and performance analysis of the instrument are introduced completely, and the cost of using the instrument should be introduced to evaluate the practicability
5.To facilitate reader comprehension, it is recommended that the authors improve the quality of the manuscript's writing.
Author Response
Thank you for the detailed review of our manuscript. We greatly appreciate the time taken to review the paper and provide feedback. We have made the following revisions to the manuscript to address your comments and suggestions:
- We agree with your assessment that including images of the system’s user interface during operation would facilitate readers’ understanding with regard to this discussion (“...by decreasing reliance on specialized medical skills, this application potentiates the ability for non-neurosurgeons to perform emergency lifesaving procedures, such as EVD placement, where neurosurgeons may not be available…”). Therefore, given the length of the current manuscript, we decided to remove this paragraph from the discussion (section 4.4 Advantages of the proposed method) in order to maintain the focus of the manuscript as an assessment of the performance of the system.
- Plots showing instrument position variability (IPV) and instrument trajectory angle variability (IAV) were included in a new figure (Figure 7) in section 3.4 Cadaveric experimental results. We used an interval plot to maintain uniformity with how data is presented in previous figures.
- The investigation into the effects of motion conditions on accuracy were expanded by including the results of motion testing under a representative ‘worst-case’ test condition. Changes are reflected in the methods (section 2.5 Motion conditions), results (section 3.2 Target registration error and trajectory angle error during movement), and discussion (section 4.2 Registration accuracy during movement).
- To address your suggestion around evaluating the cost and practicability of the system, we’ve elaborated on how the system can impact healthcare costs in section 4.4 Advantages of the proposed method.
- Writing improvements were made throughout the manuscript to improve comprehension.
Thank you again for your detailed response to our manuscript. I hope these revisions address your comments and suggestions well.
Best regards,
Isabella Chiurillo
Reviewer 2 Report
Comments and Suggestions for Authors
Thank you for the opportunity to review this manuscript.
Here, the authors present a promising solution to a well-defined problem in neurosurgery. The positive results from the assessment suggest that the Zeta Cranial Navigation System has the potential to address the limitations of traditional neuro-navigation systems in emergent settings.
Could the authors elaborate on potential drawbacks or pitfalls of the described device?
Author Response
Thank you for your detailed review of our manuscript. We greatly appreciate the time taken to review the paper and provide feedback. We have made the following revision to the manuscript to address your specific comment: In response to your suggestion, we’ve further elaborated on a potential pitfall of the system in section 4.4 Advantages of the proposed method.
I hope these revisions address your comments well. Thank you again for reviewing our manuscript.
Best regards,
Isabella Chiurillo
Round 2
Reviewer 1 Report
Comments and Suggestions for Authors
Accept